# Trained Immunity Based-Vaccines as a Prophylactic Strategy in Common Variable Immunodeficiency. A Proof of Concept Study

**DOI:** 10.3390/biomedicines8070203

**Published:** 2020-07-09

**Authors:** Kissy Guevara-Hoyer, Paula Saz-Leal, Carmen M. Diez-Rivero, Juliana Ochoa-Grullón, Miguel Fernández-Arquero, Rebeca Pérez de Diego, Silvia Sánchez-Ramón

**Affiliations:** 1Department of Immunology, IML and IdSSC, Hospital Clínico San Carlos, SN 28040 Madrid, Spain; kissgh@gmail.com (K.G.-H.); yuliochoa0887@gmail.com (J.O.-G.); mfarquero@salud.madrid.org (M.F.-A.); 2Department of Immunology, Ophthalmology and ENT, School of Medicine, Complutense University, 28040 Madrid, Spain; 3Immunodeficiency Interdepartmental Group (GIID), 28040 Madrid, Spain; rebeca.perez@idipaz.es; 4Inmunotek S.L., Alcalá de Henares, 28805 Madrid, Spain; psaz@inmunotek.com (P.S.-L.); cmdiez@inmunotek.com (C.M.D.-R.); 5Laboratory of Immunogenetics of Human Diseases, IdiPAZ Institute for Health Research, 28029 Madrid, Spain

**Keywords:** prophylaxis, TIbV, CVID, quality of life, MV130

## Abstract

Background. A major concern in the care of common variable immunodeficiency (CVID) patients is the persistence of subclinical or recurrent respiratory tract infections (RRTI) despite adequate trough IgG levels, which impacts the quality of life (QoL) and morbidity. Therefore, the development of new approaches to prevent and treat infection, especially RRTI, is necessary. Objectives. We conducted a clinical observational study from May, 2016 to December, 2017 in 20 CVID patients; ten of these patients had a history of RRTI and received the polybacterial preparation MV130, a trained immunity-based vaccine (TIbV) to assess its impact on their QoL and prognosis. Methods. Subjects with RRTI received MV130 for 3 months and were followed up to 12 months after initiation of the treatment. The primary endpoint was a reduction in RRTI at the end of the study. We analyzed the pharmacoeconomic impact on the RRTI group before and after immunotherapy by estimating the direct and indirect costs, and assessed CVID-QoL and cytokine profile. Specific antibody responses to the bacteria contained in MV130 were measured. Results. The RRTI-group treated with TIbV MV130 showed a significant decrease in infection rate (*p* = 0.006) throughout the 12 months after initiation of the treatment. A decrease in antibiotic use and unscheduled outpatient visits was observed (*p* = 0.005 and *p* = 0.002, respectively). Significant increases in anti-pneumococcus and anti-MV130 IgA antibodies (*p* = 0.039 both) were detected after 12 months of MV130. Regarding the CVID QoL questionnaire, an overall decrease in the score by more than 50% was observed (*p* < 0.05) which demonstrated that patients experienced an improvement in their QoL. The pharmacoeconomic analysis showed that the real annual direct costs decreased up to 4 times per patient with the prophylactic intervention (*p* = 0.005). Conclusion. The sublingual administration of the TIbV MV130 significantly reduced the rate of respiratory infections, antibiotic use and unscheduled visits, while increasing specific IgA responses in CVID patients. Additionally, the CVID population felt that their QoL was improved, and a decrease in expenses derived from health care was predicted.

## 1. Introduction

Common variable immunodeficiency (CVID) is one of the most common symptomatic and heterogenous primary immune disorders (PID) [1,2,3,4]. The hallmark of the disease is a defect in specific antibody production and hypogammaglobulinemia of at least two isotypes (IgG and IgA/IgM). The pathogenesis of CVID remains unknown for most patients, with 15% of cases pointing to T-B co-stimulation and BCR-dependent gene lesions [2,5,6,7]. Over 90% of CVID patients suffer from increased susceptibility to infections by different pathogens that affect several systems [1,8,9]. Additionally, the prevalence of bacterial infections might be falsely decreased due to antibiotic prophylaxis in patients with specific conditions [10]. Recurrent respiratory tract infections (RRTI) are very frequent in CVID [1,11]. Recurring infections and subclinical infection are common despite adequate immunoglobulin replacement therapy (IgRT) and can predispose patients to develop chronic lung diseases, like bronchiectasis, interstitial lung disease, granulomatous inflammation, and obstructive/restrictive lung disease, which are a leading cause of both morbidity and mortality in CVID patients [12,13,14,15,16].

Antibiotic prophylaxis in patients with CVID has been shown to improve lung function, reduce the risk of infection-related hospitalization and improve patients’ QoL in a recent double-blind placebo-controlled trial [17,18,19]. However, the risk of bacterial colonization, development of complicated pneumonia, invasive infections or infections by atypical microorganisms was not fully studied [17,18,20]. Antimicrobial resistance (AMR) represents a significant health concern worldwide. It is unclear whether PID patients are more susceptible to developing AMR [17,20,21], even though the use of long-term antibiotic prophylaxis is common in the management of PID patients. Indeed, prolonged administration of antibiotics can result in AMR to Gram positive and Gram negative pathogens, such as Group B streptococcus or *Klebsiella pneumoniae*, respectively [21]. In addition, antibiotics have limitations against several infections, mainly due to viruses, are detrimental to the already altered microbiota in PID, and are not free of adverse effects in the long-term. In this setting, complementary or alternative strategies for preventing infections in this highly susceptible population is a priority.

Immunotherapy with trained-immunity based vaccines (TIbV) are meant to enhance innate cross-protection to heterologous pathogens and to induce more efficient adaptive responses against the specific pathogens contained in the vaccine [22,23,24]. Their use in PID has not been previously reported. Sublingual administration of fully inactivated bacteria has been proposed as a safe and clinically beneficial strategy of preventing upper and lower respiratory tract infections [23,25,26], although it requires continued, daily administration. MV130 is a TIbV formulated with a combination of whole cell inactivated Gram-positive (90%) and Gram-negative bacteria (10%) that has previously been demonstrated to be a safe and effective immunotherapy against RRTI and induces long-lasting protection [23,25,26]. Bacterial preparations have been shown to elicit systemic and mucosal antigen-specific humoral and cell-mediated immunity [22,27,28,29,30]. Sublingual immunization with MV130 has been shown to up-regulate Th1, Th17 and IL-10 antigen-specific, as well as non-specific memory CD4^+^ T cell responses in vitro and ex vivo [25,26]. Concomitant up-regulation of IL-10 by MV130 may add immunomodulatory potential in the setting of autoimmune or inflammatory imbalance in CVID [25]. Finally, a recent clinical trial in children with recurrent wheezing, a disease mostly triggered by respiratory viruses, demonstrated that MV130 significantly decreased wheezing attacks in children. MV130 induced trained immunity and elicited an antiviral effect in experimental models with vaccinia and flu viruses [31,32].

We used a proof-of-concept study to determine the beneficial effects of a novel adjuvant therapeutic strategy with a TIbV in CVID patients suffering RRTI despite adequate trough IgG levels and antibiotic prophylaxis.

## 2. Methods

The present study is a single-institution retrospective observational study conducted at the Clinical Immunology Department of the Hospital Clínico San Carlos, Madrid, Spain, from May 2016 to December 2017.

Approval for the study was obtained from the Hospital Clinico San Carlos institutional research Ethics Committee (16/511-E). Written informed consent was obtained from all patients for inclusion in the study protocol.

### 2.1. Population

Data from a cohort of 20 patients with a definite diagnosis of CVID, aged between 18 to 65 years, and serum IgG above 600 mg/dL on IVIg therapy were recorded. The exclusion criteria were: patients with lymphoproliferative disorders, asthma treated with inhaled or systemic corticosteroids, inability or unwillingness to provide written informed consent or significant medical or psychiatric illness that, in the opinion of the treating clinician, precluded participation.

Subsequently, the cohort was subdivided into two groups according to whether or not the patient presented with RRTI, which was defined by ≥3 episodes of upper (rhinitis, sinusitis, otitis, pharyngitis, tonsillitis) or lower respiratory infection (bronchitis), ≥2 episodes of severe respiratory infection (pneumonia), or the need for antibiotics for almost two months/year in the previous 12 months as described elsewhere [9,10]. This group was referred to as the RRTI-Group. According to our clinical routine protocols at the time, the RRTI-group (n = 10) had received a TIbV MV130 (Bactek^®^, Inmunotek, Madrid, Spain) daily for 3 months. According to the drug data sheet, the MV130 sublingual immunotherapy was indicated for recurrent infections. MV130 is composed of whole complete inactivated bacteria, as follows: 90% Gram-positive bacteria such as *Staphylococcus* spp., *Streptococcus pneumoniae*, and 10% Gram-negative bacteria such as *Klebsiella pneumoniae*, *Moraxella catarrhalis*, *Haemophilus influenzae*.

As part of our regular medical follow-up, RRTI-group was evaluated before and at 12 months of sublingual prophylaxis, with a complete medical history, physical exam and immunological profile. The sera were stored at −80 ºC. The remaining patients (n = 10) that did not fulfil the RRTI-Group criteria were designated as the Non-RRTI-Group. This group was considered as a control group for the comparison of the immunological CVID profile (Figure 1).

### 2.2. Immunological Assessment

We measured the presence of specific serum IgA and IgG against MV130 and *S. pneumoniae* (the main Gram- positive component) in the RRTI group by ELISA. The samples were stored frozen at −80 °C before processing. Briefly, 96-well non-tissue culture-treated plates were pretreated with 100 µL of poly-L-lysine (stock at 0.01%, 1:1000 dilution) (Sigma-Aldrich) for 1 h under UV light and coated with the appropriate whole-cell heat-inactivated bacteria or polybacterial mixture (300 nephelometric turbidity units (NTU), ~10^9^ bacteria) overnight at 4 °C, and subsequently incubated with human sera for 2 h at room temperature. IgA and IgG antibodies were detected using the following reagents: biotin rat anti-human IgA or IgG (both from Sigma-Aldrich) and streptavidin horseradish peroxidase (HRP) (Sigma-Aldrich). Peroxidase activity was revealed by the addition of o-phenylenediamine dihydrochloride (Sigma-Aldrich) and the reaction was stopped with HCl 1N. Plates were read on an ELISA reader at 490 nm (Triturus Elisa, Grifols). 

Likewise, an extensive panel of Th1/Th2/Th17/Treg cytokine profiles, growth factor, and chemokines was carried out (Pro human cytokine 27-plex kit, Bio-Plex™ from BioRad Inc) at baseline and after treatment (12 months) in the RRTI group and was compared with the Non-RRTI-Group in sera samples, following the manufacturer’s instructions. The samples were stored frozen at −80 °C before analysis.

### 2.3. Endpoints

The primary endpoint of the study was to evaluate the percentage of CVID patients on MV130 who had reduced RRTI in terms of frequency of infections, as well as safety issues. The secondary endpoints were the reduction in the number of antibiotic cycles and in the unscheduled outpatient visits due to infections.

Each CVID subject in the RRTI-group was given the CVID-QoL assessment, before and 12 months after treatment. This was measured by the Health-Related Quality of Life questionnaire for adults with common variable immunodeficiency proposed by Quinti et al. [33]. The questionnaire comprises 32 items with responses given on a 4-point scale, with 0 = “never” and 4 = “always,” with higher values generally indicating increasing disability (range, 0 to 160 points). 

We evaluated the pharmacoeconomic impact of this prophylactic approach by comparing the characteristics of the RRTI-group before and after vaccination measured by direct and indirect costs. The direct costs estimation was comprised of the medical direct costs and non-medical direct costs. The medical direct costs included those costs attributable to health care visits for the treatment of the disease including primary and secondary care consultations, prescriptions, hospital-based procedures, nature and length of inpatient stay, surgery, and over-the-counter purchases by patients [34]. Non-medical direct costs included travel expenses, home care, etc. The information was collected from the CRF (patient self-reported information) and completed with medical records. These costs were assessed using the official rates most recently published for the public health service for a follow-up period of 12 months. The direct costs were calculated according to the following formula: Direct Costs = Medical direct costs + non-medical direct costs. The evaluation of indirect costs included those derived from work suspension related to the disease. For working patients, an average hourly wage (€14.04) was applied to all absentee time, based on salary cost per productive hour, type of working day, and activity sectors in the National Statistics Institute data [35]. 

### 2.4. Statistical Analysis

The integration of all obtained data (from laboratory to clinical data) was done by using bivariate correlations and multivariate analysis in order to detect associations between different variables. Patients were stratified according to the obtained results (different clinical and immunological responses). Normal distribution and testing of outliers were analyzed by means of Shapiro–Wilk and Tukey’s range tests, respectively. Data were analyzed by Chi-squared, Fisher’s exact, Pearson, and Spearman correlation coefficient, Mann Whitney U-tests, paired Student’s *t* test and Wilcoxon signed-rank test, and one-sample *t* test using SPSS (Chicago, Illinois) and GraphPad Prism software (GraphPad Software, La Jolla, CA, USA). A p-value below 0.05 was considered as statistically significant.

## 3. Results

### 3.1. MV130 Significantly Decreased Respiratory Infection Rate, Antibiotic Use and Unscheduled Outpatients’ Visits in CVID Patients

All patients suffering from RRTI completed the daily treatment with MV130 for 3 months and were followed up for 12 months. The RRTI-Group treated with TIbV MV130 significantly decreased the median (range min-max) infection rate from 3.00 (1–7) to 0.00 (0–2) (*p* = 0.006) post-treatment. All of the patients had a reduced infection rate in the average follow-up time of 12 months (Figure 2). Thirty percent maintained upper respiratory tract infections. Antibiotic consumption significantly decreased from 5.00 cycles (3–7) to 1.00 (0–1) cycle (*p* = 0.005); the number of infectious-related unscheduled outpatient visits significantly declined from 5.00 (2–6) to 1.00 (0–3) (*p* = 0.002); and work absenteeism decreased from 2.00 (0–3) to 0.00 (0–2) days (*p* = 0.005) in the 12 months after initiation of MV130 administration (Figure 3, Appendix A). None of the patients reported any side effects regarding MV130, either local at the site of administration or systemic and no adverse effects were noted during the 12-month observational period.

### 3.2. Immune Profile

As a preliminary approach, to assess whether sublingual immunotherapy with MV130 triggers a systemic humoral response as well, blood samples were assayed for specific IgG and IgA at baseline and after 12-months of initiating MV130. A significant increase in both anti-*S. pneumoniae* and anti-MV130 IgA (*p* = 0.039) but not IgG (*p* = 0.094) serum antibodies following MV130 immunization were observed (Figure 4). As *S. pneumoniae* represents the main Gram-positive component in MV130, these results point to the induction of a humoral specific response upon bacterial immunotherapy. 

We then analyzed the baseline expression of an extensive panel of serum cytokines, chemokines and growth factors in both cohorts (RRTI-group and Non-RRTI group). The RRTI-group showed higher baseline expression of IL-4, IL-17, IFN-γ, MCP-1 (MCAF) and TNF-α, with respect to Non-RRTI group, without statistical differences. Interestingly, when comparing the RRTI-group at baseline with respect to 12 months after MV130, a significantly lower concentration of serum IL-17 (49.49; 0–90.04 vs. 26.85; 0–70.27, *p* < 0.05) with a significant fold reduction in serum IL-17 (0.649, *p* < 0.05); and IL-4 (23.01; 7.03–27.44 vs. 16.92; 5.13–26.35, *p* < 0.05), and fold reduction of IL-4 (0.780, *p* < 0.05) were observed. This decrease in serum IL-17 restored cytokine levels to those found in non-RRTI individuals (data not shown). The cytokine expression measured pre- and post-immunization in the RRTI-group can be seen in Figure 5. 

### 3.3. Perceived QoL Improved in CVID Patients after MV130

We evaluated the self-reported quality of life of each RRTI-group patient at baseline and 12 months after receiving prophylaxis with MV130 based on the adapted CVID-QoL questionnaire proposed by Quinti et al. [33]. An overall significant decrease in score was observed (the median (range) of the global score decreased from 47.0 (27–86) to 39.5 (13–86), *p* < 0.05), reflecting an improvement in QoL for patients (Appendix A). The median percentage of pre to post MV130 CVID-QoL improvement was 16.87% (range, 0% to 51.85%). Only one item, “Run out of medications” showed an increase in perception post-treatment while the item that decreased the most was “Difficulty in usual activities”.

### 3.4. Pharmacoeconomic Impact

The economic impact of this prophylactic approach (pre/post-MV130 immunization) was evaluated in terms of direct costs, which were comprised of medical and non-medical direct costs. The indirect costs evaluation included those derived from work suspension related to the disease (Appendix A).

Regarding the direct costs, the total average cost for ambulatory care for patients with RRTI is 1656 €/patient, according to the official report of the Spanish Ministry of Health, Consumer Affairs and Social Welfare in 2017 [36]. This amount can increase to 3962 €/patient when hospital admission is required. The estimated real median annual direct cost is approximately 18,600 €/patient in the RRTI group (Table 1). Taking into account the cost of the intervention, our results showed that the real annual median direct cost decreases up to 4-fold per patient with the prophylaxis intervention of TIbV MV130 (from €18,600 to €4500, *p* < 0.005).

With regard to indirect costs, 14 euros per hour was defined as the cost of absenteeism due to illness, as well as presenteeism due to the reduction in the percentage of productivity [36]. A median of 2 days of absenteeism (the daily workday is 8 h), equivalent to €28, added up to €308 for a cumulative 22 h of labor absenteeism, giving a total of 336 €/patient of annual indirect costs in the pre-immunization RRTI group. These costs are only imperceptibly different (median work absenteeism post-immunization was 0 days) to those calculated after the prophylaxis intervention of MV130 (*p* < 0.005).

## 4. Discussion

Subclinical and recurrent infections remain a considerable healthcare concern in a subgroup of CVID patients, and they affect the risk of complications and hence the prognosis [10,37]. Bronchiectasis is a consequence of recurrent lower respiratory tract infections and the most common chronic pulmonary complication at CVID patients. It is also the most important cause of morbidity in these patients [11,38,39]. The lung damage could contribute to the high recurrence of infections in a specific group of CVID patients. To our knowledge, this is the first study that shows the beneficial effects of an adjuvant TIbV on the control of recurrent upper respiratory tract infections in CVID patients. 

IgRT has been demonstrated to be the best intervention to change the course of CVID by preventing infections [40]. Still, despite adequate serum trough IgG levels, some CVID patients on IVIg/SCIg replacement suffer from subclinical infections and RRTI that require constant or frequent antibiotics and/or they may develop bronchiectasis, which negatively impacts their vital prognosis [40,41,42]. Therefore, new adjuvant prophylactic approaches to infection, especially in this CVID group are greatly needed.

Most pathogens affecting CVID patients enter the body by the oropharyngeal mucosa. In this proof-of concept observational study, MV130 significantly decreased the rate of respiratory tract infections, the frequency of antibiotic consumption and the frequency of unscheduled outpatient visits. Moreover, specific IgA responses were observed despite the intrinsic antibody defect in CVID. IgA secretion has been strongly associated with decreased rates of respiratory infections, and thus overall mortality risk [43,44]. Significantly higher specific anti-pneumococcal and anti-MV130 IgA antibodies is highly relevant in the context of CVID, especially because these patients are usually low antibody producers. Specific, significant differences in serum IgA may be explained by the mucosal route of administration of the treatment. MV130 has been shown to increase mucosal IgA secretion [45]. IgA synthesis occurs in mucosal-associated lymphoid follicles through B cell hypermutation and class switch recombination, and is secreted by plasma cells. Secretory IgA (sIgA) is exclusively present on mucosal surfaces and consists of dimeric IgA linked via the J-chain to the secretory components [46,47]. Circulating sIgA constitutes a natural key systemic anti-inflammatory factor due to its ability to interact with DCs through receptors such as ICAM-3, DC-SIGN, favoring the generation of Ag-specific regulatory CD4^+^ T cells [46,48], which further suggests that IgA has a homeostatic role between commensal microorganisms and pathogens.

Even though the main strength of MV130 as a TIbV may rely on the induction of broad-spectrum, non-specific protection, the data support the hypothesis that specific adaptive responses are also generated that may also correlate with the clinical benefits that result from MV130 administration. 

The MV130 polybacterial mixture is prepared by using whole complete inactivated bacteria (90% Gram positive and 10% Gram negative bacteria). This formula was initially standardized according to the ad hoc prevalence of isolated pathogens for the preparation of individualized vaccines for patients with recurrent respiratory tract infections. The Gram positive and Gram negative bacteria act in a synergistic and complementary way in the activation of innate immunity [25]. MV130 has been shown to activate trained immunity in vitro and ex vivo [26,32] by inducing epigenetic and metabolic changes. Regarding the route of administration, mucosal routes for vaccines against respiratory pathogens have been shown to be effective and safe compared to other routes [26,47,48,49,50,51], and they induce a greater immune response in the respiratory tract mucosa. 

MV130 may enhance innate immunity at the entry portal of the pathogen. Several studies have highlighted the important role of mucosal preparations formulated with inactivated bacteria in controlling respiratory exacerbations in the setting of chronic obstructive pulmonary disease [52] and preliminary results in patients with RRTI without a defined PID [31,51]. This strategy might provide a potential solution to clinical problems not fully covered by other treatments.

Trained immunity is based on biological processes derived from innate immune memory associated with intracellular signals driving deep metabolic changes and epigenetic modifications that result in reprogramming in cells of innate immunity [53,54]. These effects occur synergistically where various metabolites and metabolic enzymes act as cofactors in the process of epigenetic modification such as transcriptional changes, activation/inhibition of histones, or modulation of cytokine expression [55]. Several therapeutic strategies have been designed to modulate trained immunity [23,24,25,26,56]. TIbVs have the potential to improve immune responses by protecting against secondary related or bystander infections, and reversing immunotolerance states as well as chronic inflammatory conditions [22,24,27,29,51].

The main goal of this proof-of-concept study is to generate preliminary data on the beneficial effects and potential risks of these polybacterial mucosal vaccines in order to support the design a larger randomized double-blind clinical trial. The present study has provided an initial evaluation of the safety of the intervention in the target population, which is a major clinical endpoint. The main weakness of this study is the observational retrospective design, which may result in missing variables not recorded in routine clinical practice. However, due to the promising results obtained, this study could help in the design of further studies. 

MV130 induces the activation of TLR and NLR signaling pathways in human dendritic cells (DCs), which secrete cytokines such as TNF-α, IL-6 and IL-1β related to trained immunity. These DCs also possess the capacity to promote Th1 and Th17 responses through the RIPK2 and MyD88 signaling pathway under the control of IL-10. In addition, DCs have a large number of pattern recognition receptors that facilitate the synergistic action of the innate and adaptive immune systems [25].

Several studies have identified abnormalities in cytokine secretion in CVID patients, including decreased production of IFN-γ, IL-2, IL-5, IL-7, IL 4, IL-10 or IL-12 [57,58,59,60,61,62]. IL-10 is essential to avoid excessive deleterious responses, to enhance pathogen clearance, and to keep tissue homeostasis [6,63,64,65,66]. No significant changes in serum IL-1β or IL-10 with MV130 were observed. However, we consider that this may be because the cytokine measurement was performed 12-months after MV130 administration, and thus, is reflects the long-term regulation of the proinflammatory status due to the decrease in recurrent infections in this group of patients. The high baseline expression of IL-17 observed in our cohort has already been described in previous studies [58]. However, their clinical significance and the correlation with other biomarkers such as IFN-γ tend to be heterogeneous. Nevertheless, the correct way to observe the real behavior and significance of these results is to carry out in vitro activation of peripheral blood cells of our patients that is stimulated by various ligands, and measure the different expression of cytokines during and at the end of the treatment.

QoL is a multidimensional concept that encompasses several elements such as the physical, psychological and social well-being of individuals. The main point of QoL is to assess an individual’s perception of the impact of illness on his/her life at a similar level to that of the clinical factors in the disease’s prognosis [67]. RRTI has a critical effect on the QoL of CVID patients. The prophylaxis intervention of TIbV MV130 improved the global perception of QoL in more than 50% of the CVID patients (*p* < 0.005). “Difficulty in usual activities” was the component with the highest post treatment decrease in score. The only element that scored higher after the prophylactic intervention was “Run out of medications”. These findings further highlight the positive impact of the prophylaxis with a TIbV such as MV130 on the QoL of CVID patients. 

CVID treatment without complications requires IgRT, which implies an annual cost of about 15,700 €/patient. In selected patients, treatment costs may be controlled by modifying the dosage of IgRT, changing the administration route to SCIg or changing the intervals between administrations [68]. The reduced frequency of infections among CVID patients after treatment with IVIg/SCIg translates into health care cost savings [69]. However, a subgroup of CVID patients is unable to improve their infection profile in spite of IVIg/SCIg treatment, consequently, the implementation of an effective alternative treatment can have a direct impact on the abovementioned health care costs of such patients. 

Despite the limitations inherent to this observational small study, we show that the clinical benefits of intervention by TIbV are accompanied by healthcare cost savings of approximately 9000–14,000 €/patient per year in terms of hospital, healthcare and lost wages. Considering the high cumulative cost of the treatment of this disease, bacterial immune-stimulation could be an effective strategy to control subclinical infection and respiratory tract exacerbations, and to reduce costs in CVID patients.

## 5. Conclusions

There is a need to develop alternative or adjuvant strategies to prevent infections in selected CVID patients with subclinical and/or recurrent infections. The preliminary data in our small cohort of CVID patients with recurrent infections despite adequate trough IgG levels show that mucosal vaccines are a potentially useful strategy for preventing infections at the actual point of entry for most pathogens. The study shows enough promise to be repeated in a double-blind clinical trial. The prophylaxis with sublingual TIbV MV130 based on whole cell inactivated bacteria resulted in significant clinical benefits in terms of recurrence and severity of respiratory tract infections, with a concomitant improvement in QoL perception as well as a decrease in health care expenses.

## Figures and Tables

**Figure 1 biomedicines-08-00203-f001:**
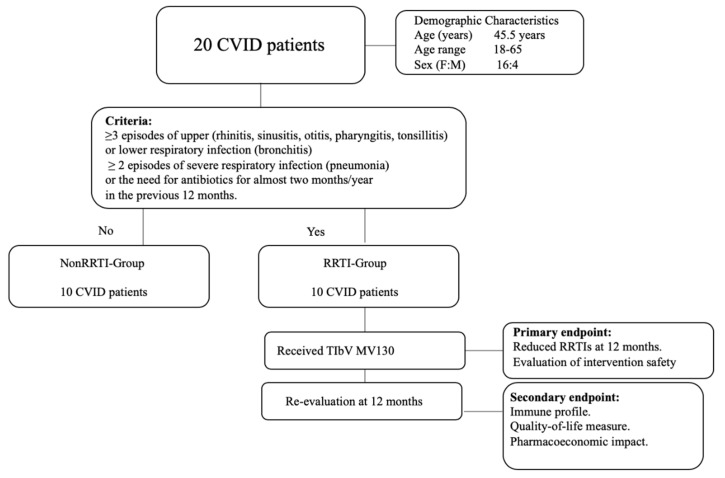
Flow chart of the classification of common variable immunodeficiency (CVID) patients according to clinical criteria.

**Figure 2 biomedicines-08-00203-f002:**
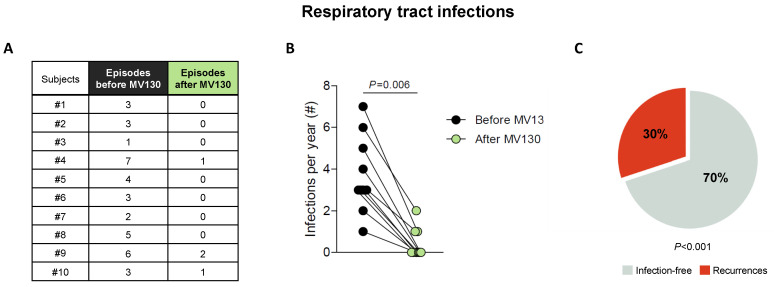
MV130 significantly reduces the incidence of respiratory infections (**A**–**C**). (**A**,**B**) Number of respiratory tract infectious episodes scored 1 year prior to immunization (black) and in the 12 months after the initiation of immunotherapy with MV130 (green). (**C**) Percentage of subjects that remained free of infection (grey) or suffered recurrences (red) in the 12 months following MV130 administration. Data from 10 subjects are shown. (**B**) Lines link paired values. Normal distribution was evaluated using the Shapiro–Wilk test, *p* value was calculated using Wilcoxon signed-rank test. (**C**) *p* value was calculated using Fischer’s exact test, compared with rates prior to MV130 initiation (100% of subjects suffering infections).

**Figure 3 biomedicines-08-00203-f003:**
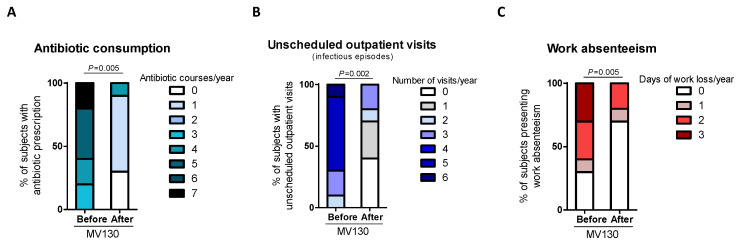
Prophylaxis with MV130 significantly decreases the rate of healthcare resources consumption and work absenteeism. (**A**–**C**) Antibiotic consumption (**A**), visits to emergency unit (**B**) and work absenteeism (**C**) during the year before and after the initiation of MV130 treatment. Bars show the relative number of antibiotic courses (**A**), emergency unit visits (**B**) or days of work lost (**C**) in the total of subjects recorded. Data from 10 subjects are shown. Normal distribution was evaluated using the Shapiro–Wilk test. *p* values were calculated using Wilcoxon signed-rank test.

**Figure 4 biomedicines-08-00203-f004:**
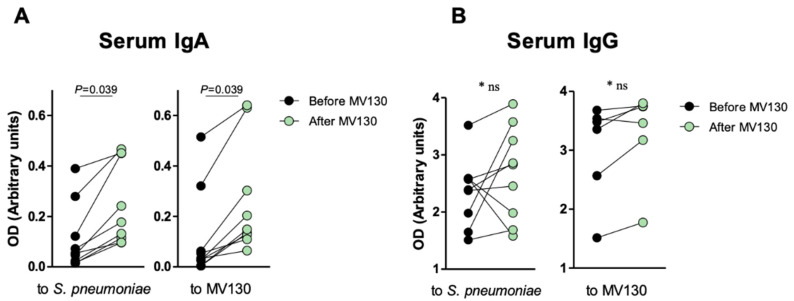
Specific anti-MV130 IgA and IgG antibodies. Prophylaxis with MV130 increases serum IgA antibody production. (**A**,**B**) Serum IgA (**A**) and IgG (**B**) antibodies against *S. pneumoniae* (left panels) or the bacterial mixture (right panels), collected from subjects before and after 12 months following MV130 immunotherapy analyzed by ELISA. Data from 6–10 individuals are shown. Lines show paired values. Normal distribution was evaluated using the Shapiro–Wilk test. *p* values were calculated using Paired Student’s *t*-test or Wilcoxon signed-rank test.

**Figure 5 biomedicines-08-00203-f005:**
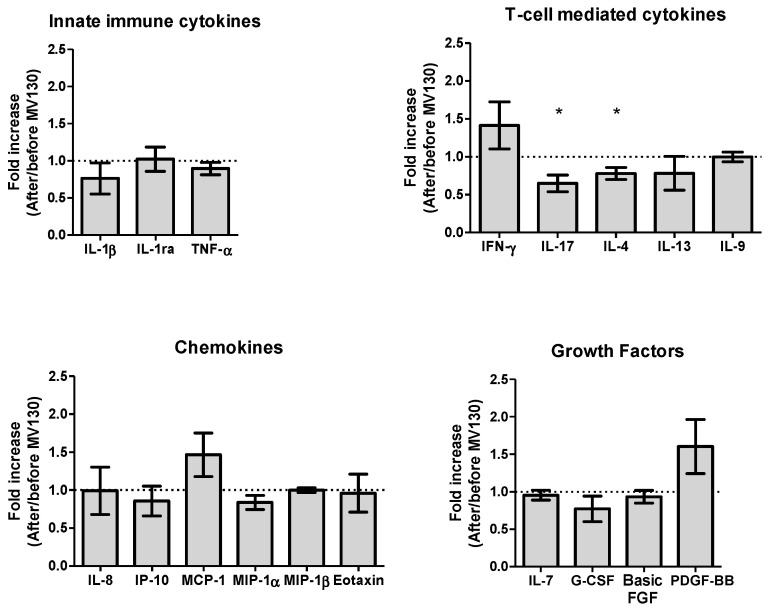
Prophylaxis with MV130 modulates serum cytokine, chemokine and growth factor secretion pattern. Cytokines (upper panels), chemokines (left bottom panel) and growth factors (right bottom panel) secreted in serum one year following MV130 administration determined by Luminex technique. Fold induction relative to basal level (before MV130 treatment) is shown. Undetectable values were found for IL-2, IL-5, IL-10, IL-12(p70), IL-15, GM-CSF, RANTES and VEGF. Data from 9–10 subjects are shown as mean ± SEM of fold increase. Outliers were identified by means of Tukey’s range test on represented values. Normal distribution was evaluated using the Shapiro–Wilk test. *p* values were calculated using one-sample *t*-test with a theoretical value of 1 (no fold induction). * *p* < 0.05.

**Table 1 biomedicines-08-00203-t001:** Costs of the most frequent conditions affecting CVID patients with RRTI—comparison of the year before and the year after the prophylaxis with TIbV (Euros). (#: number)

Condition	Average # of Episodes before TIbV MV130	Average # of Episodes after TIbV MV130	Cost per Patient per Episode/Day €	Annual Cost per Patient before TIbV MV130 €	Annual Cost per Patient after TIbV MV130 €	Annual Savings per Patient with TIbV MV130 €
# of RRTIs	3.7	0.4	1656	6127	662	5464
# of physician/hospital/ER visits	4.4	1.1	1288	5667	1416	4250
# Days Hospitalizations for RRTIs	7	3	792	5546	2377	3169
Cycles of antibiotics	4.8	1	259	1243	259	984
School/work days missed (Absenteeism)	1.6	0.5	14	22.4	7	15
Total per patient				18,606	4722	13,884
Annual cost TibV MV130 prophylaxis					190	13,694

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
