# Peer review of "Trained Immunity Based-Vaccines as a Prophylactic Strategy in Common Variable Immunodeficiency. A Proof of Concept Study"

_biomedicines, 2020, doi:10.3390/biomedicines8070203_

Round 1

Reviewer 1 Report

The present manuscript discusses the positive effect of the polybacterial preparation MV130 on the number of infections, quality of life, the level of specific immunoglobulins and the level of cytokines in CVID patients with an increased number of infections. The topic of the article is current. The results are well processed, the discussion is sufficient.
I see the only shortcoming in the presentation of statistical results in the text of the article. If the measured values ​​are processed by a parametric test, the result is reported as the mean and standard deviation. When processing values ​​with a nonparametric test, the rule is that the results are reported as medians and then min and max.
Authors of the manuscript report the results as median and IQR, while some results have the value 0 and IQR has the value one, ie for example they get to negative values ​​in the number of infections, here it is necessary to correct the form of the presented results.
In addition, it is not usual to state the values ​​of medians, etc. in the abstract. Usually, only a statistically significant decrease or increase or a maximum probability value is reported.
Even in Figure 3B, the value of P should not be given, but the phrase: is not significant, or the abbreviation ns.

In the abstract page 1, line no.23 in the word RTI another one R is missing.

Chapter Introduction, page 2, line art replace the word coestimulation with the word costimulation.

Please check the correctness and availability of citations number 21, 27, 31, 33, 44, 63, 64

Author Response

Comments and Suggestions for Authors

I see the only shortcoming in the presentation of statistical results in the text of the article. If the measured values ​​are processed by a parametric test, the result is reported as the mean and standard deviation. When processing values ​​with a nonparametric test, the rule is that the results are reported as medians and then min and max.
Authors of the manuscript report the results as median and IQR, while some results have the value 0 and IQR has the value one, ie for example they get to negative values ​​in the number of infections, here it is necessary to correct the form of the presented results.

A: We appreciate the Reviewer’s careful reading of the manuscript and constructive comments and suggestions. Based on your comment, we have corrected the presentation of statistical results in the text. We report results as medians and then min and max in the new manuscript to express the idea more clearly.

In addition, it is not usual to state the values ​​of medians, etc. in the abstract. Usually, only a statistically significant decrease or increase or a maximum probability value is reported. Even in Figure 3B, the value of P should not be given, but the phrase: is not significant, or the abbreviation ns.

A: Thanks for your observation. This aspect has been corrected in the Abstract section and in the figure 3B of the revised manuscript.

In the abstract page 1, line no.23 in the word RTI another one R is missing. Chapter Introduction, page 2, line art replace the word coestimulation with the word costimulation. Please check the correctness and availability of citations number 21, 27, 31, 33, 44, 63, 64

A: Thanks again for your comment. These aspects were corrected in the revised manuscript.

Finally, we appreciate very much all of your insightful comments. Thank you for helping us to improve our paper.

Reviewer 2 Report

The manuscript of Guevara-Hoyer et al addresses the prevention and treatment of recurrent infections (especially RTI – respiratory tract infections) in CVID patients. The authors hypothesised that regular mucosal exposure to the polybacterial preparation MV130 – a preparation for which there is a reasonable literature – may influence the QoL and prognosis of CVID patients. This was clearly identified by the authors as a small proof of concept study in which two groups of 10 patients with CVID with or without a prior history or recurrent RTI. In their study participants received the MV130 preparation for 3 months and were then followed for the next 9 months (12 months in total). As written in the manuscript the primary endoint was to measure the frequency (hopefully reduced) of RTI in terms of frequency of infections and the secondary endopoints were 1. reduction in the number of antibiotic cycles, 2 the number of unscheduled outpatients’ visits due to infections, and the pharmacoeconomic impact of MV130 exposure. The manuscript clearly describes the work done, and how the study reached the primary and secondary endpoints. Detailed mechanistic studies were not part of the endpoints, and no real data is presented, although the authors report circulating cytokine and IgA levels. Overall, the study addresses the question asked by the authors, the manuscript is well written and I have a few comments that need to be addressed before the manuscript can be considered acceptable for publication.

  1. The authors seem aware of clinical trial reporting guidelines as they have included for example Figure 5 (CONSORT diagram). They need however, to include the ethical approval information – full committee name and approval reference number
  2. Interventional clinical trials like this need to be pre-registered with the primary and secondary endpoints clearly identified and placed in a suitable repository before the study starts to avoid them being changed during the study to what actually worked – I must stress that I am not saying that the authors did this - however, it is normal practice to register the study, particularly when there is the potential conflict of interest as the authors have correctly declared. Details of study registration must be provided for this study before it can be published.
  3. The study, by design, has inherent biases – inclusion bias, cognitive bias and placebo biases – this is because it was not a double-blind study. While this was perhaps not necessary for a disease with limited patient numbers and in a small proof of concept study, this needs to be clearly addressed in the manuscript – and one of their conclusions has to be that the study shows enough promise to be repeated in a double-blind manner.
  4. The authors look at “specific anti-MV130 IgA antibodies”. Please include in the discussion a brief paragraph on the IgA, and the limitations (ie permissiveness of IgA, detecting multiple species, mucosal secretion etc..) of measuring IgA rather than IgG.

I would like, however, to thank the authors for their study, as this addresses the key points that are often overlooked in biomedical research, particularly the QoL and the number of unscheduled hospital visits.

Author Response

The authors seem aware of clinical trial reporting guidelines as they have included for example Figure 5 (CONSORT diagram). They need however, to include the ethical approval information – full committee name and approval reference number.

Interventional clinical trials like this need to be pre-registered with the primary and secondary endpoints clearly identified and placed in a suitable repository before the study starts to avoid them being changed during the study to what actually worked – I must stress that I am not saying that the authors did this - however, it is normal practice to register the study, particularly when there is the potential conflict of interest as the authors have correctly declared. Details of study registration must be provided for this study before it can be published.

  1. Thank you very much for raising this important point. The study is an exploratory observational work based on routine clinical practice in our unit (proof of concept study) to initially test the perception of benefit of these polybacterial preparations in the setting of CVID. MV130 sublingual vaccine was indicated for recurrent infections (according to drug data sheet), authorized by the Spanish Agency and financial support of the National Health System. We presented the study to the Hospital Clinico San Carlos institutional research Ethics Committee as an observational retrospective study. We have added and corrected this aspects in the Methods section of the revised manuscript. (Page. 2 88-94).

“The present study is a single-institution retrospective observational study, conducted at the Clinical Immunology Department, of the Hospital Clínico San Carlos, Madrid, Spain, within the period from May, 2016 to December, 2017.

Approval for the study was obtained from the Hospital Clinico San Carlos institutional research Ethics Committee (16/511-E). Written informed consent was obtained from all patients for inclusion in the study protocol”.

The study, by design, has inherent biases – inclusion bias, cognitive bias and placebo biases – this is because it was not a double-blind study. While this was perhaps not necessary for a disease with limited patient numbers and in a small proof of concept study, this needs to be clearly addressed in the manuscript – and one of their conclusions has to be that the study shows enough promise to be repeated in a double-blind manner.

  1. Thanks for your observation, we obviously agree. We have extended a paragraph on the limitations of our study the Discussion and Conclusion sections of the revised manuscript, as required by this Reviewer.

(Page. 9 Line. 324-330).

“The main goal of this proof-of-concept study is to generate preliminary data on the beneficial effects and potential risks of these polybacterial mucosal vaccines in order to support the design a larger randomized double-blind clinical trial. The present study has provided an initial evaluation of intervention safety in the target population, which is a major clinical endpoint. The main weaknesses of this study is the observational retrospective design, which may have lost variables not recorded in routine clinical practice. However, due to the promising results obtained, it would help in the design of further studies”.

(Page. 10 Line. 374-377).

“The preliminary data in our small cohort of CVID patients with recurrent infections despite adequate trough IgG levels show that mucosal vaccines are a potentially useful strategy for preventing infections at the actual point of entry for most pathogens. The study shows enough promise to be repeated in a double-blind clinical trial”.

The authors look at “specific anti-MV130 IgA antibodies”. Please include in the discussion a brief paragraph on the IgA, and the limitations (ie permissiveness of IgA, detecting multiple species, mucosal secretion etc..) of measuring IgA rather than IgG.

I would like, however, to thank the authors for their study, as this addresses the key points that are often overlooked in biomedical research, particularly the QoL and the number of unscheduled hospital visits.

A: Thank you. Following the Reviewer’s comments, we have added a paragraph discussing the relevant role of IgA specific responses given the mucosal route administration of the vaccine; and the dual role of IgA at mucosal sites both as effector response and anti-inflammatory, which may influence immune mucosal environment in CVID. We evaluated specific anti-MV130 IgA and IgG antibodies.

(Page. 8 Line. 286-296).

“Significantly higher specific anti-pneumococcal and anti-MV130 IgA antibodies is of outstanding relevance in the setting of CVID, especially considering these patients are usually low antibody producers. Specific serum IgA significant differences may be explained via the mucosal route of administration of the vaccine. MV130 has shown to increase mucosal IgA secretion [43]. IgA synthesis occur at mucosal-associated lymphoid follicles through B cell hypermutation and class switch recombination, and secreted by plasma cells. Secretory IgA (sIgA) is exclusively present at mucosal surfaces and consists of dimeric IgA linked via the J-chain to the secretory components [44,45]. Circulating sIgA constitutes a natural key systemic anti-inflammatory factor through its ability to interact with DCs through receptors such as ICAM-3, DC-SIGN, favoring the generation of Ag-specific regulatory CD4+ T cells [44,46], further suggesting a homeostatic role of IgA between commensal microorganisms and pathogens”.

Again, we sincerely appreciate your careful reading of the manuscript and all the constructive comments and suggestions that have greatly improved our work. We hope that our revision meets your approval.

Round 2

Reviewer 2 Report

The authors have adequately replied to my concerns